# Pregnancy Outcomes and Maternal Insulin Sensitivity: Design and Rationale of a Multi-Center Longitudinal Study in Mother and Offspring (PROMIS)

**DOI:** 10.3390/jcm10050976

**Published:** 2021-03-02

**Authors:** Anoush Kdekian, Maaike Sietzema, Sicco A. Scherjon, Helen Lutgers, Eline M. van der Beek

**Affiliations:** 1Laboratory of Pediatrics, University Medical Centre Groningen, University of Groningen, Hanzeplein 1, 9713 GZ Groningen, The Netherlands; A.Kdekian@umcg.nl; 2Clinical Endocrinology, Medical Centre Leeuwarden, Henri Dunantweg 2, 8934 AD Leeuwarden, The Netherlands; m.sietzema@umcg.nl (M.S.); Helen.lutgers@mcl.nl (H.L.); 3Department of Gynaecology and Obstectrics, University Medical Centre Groningen, Hanzeplein 1, 9713 GZ Groningen, The Netherlands; s.a.scherjon@umcg.nl

**Keywords:** pregnancy, insulin sensitivity, (early) pregnancy, pregnancy outcomes, growth, glucose, overweight, obesity, LGA, maternal health, child health, gestational diabetes mellitus, OGTT, meal tolerance test, postprandial

## Abstract

The worldwide prevalence of overweight and obesity in women of reproductive age is rapidly increasing and a risk factor for the development of gestational diabetes (GDM). Excess adipose tissue reduces insulin sensitivity and may underlie adverse outcomes in both mother and child. The present paper describes the rationale and design of the PRegnancy Outcomes and Maternal Insulin Sensitivity (PROMIS) study, an exploratory cohort study to obtain detailed insights in insulin sensitivity and glucose metabolism during pregnancy and its relation to pregnancy outcomes including early infancy growth. We aim to recruit healthy pregnant women with a body mass index (BMI) ≥ 25 kg/m^2^ before 12 weeks of gestation in Northern Netherlands. A total of 130 woman will be checked on fasted (≤7.0 mmol/L) or random (≤11.0 mmol/L) blood glucose to exclude pregestational diabetes at inclusion. Subjects will be followed up to six months after giving birth, with a total of nine contact moments for data collection. Maternal data include postprandial measures following an oral meal tolerance test (MTT), conducted before 16 weeks and repeated around 24 weeks of gestation, followed by a standard oral glucose tolerance test before 28 weeks of gestation. The MTT is again performed around three months postpartum. Blood analysis is done for baseline and postprandial glucose and insulin, baseline lipid profile and several biomarkers of placental function. In addition, specific body circumferences, skinfold measures, and questionnaires about food intake, eating behavior, physical activity, meal test preference, mental health, and pregnancy complications will be obtained. Fetal data include assessment of growth, examined by sonography at week 28 and 32 of gestation. Neonatal and infant data consist of specific body circumferences, skinfolds, and body composition measurements, as well as questionnaires about eating behavior and complications up to 6 months after birth. The design of the PROMIS study will allow for detailed insights in the metabolic changes in the mother and their possible association with fetal and postnatal infant growth and body composition. We anticipate that the data from this cohort women with an elevated risk for the development of GDM may provide new insights to detect metabolic deviations already in early pregnancy. These data could inspire the development of new interventions that may improve the management of maternal, as well as offsrping complications from already early on in pregnancy with the aim to prevent adverse outcomes for mother and child.

## 1. Introduction

Insulin sensitivity normally decreases by the end of the second trimester due to the effect of placental hormones, such as human placental lactogen (HPL), progesterone, and human placental growth hormone, to ensure a continuous supply of nutrients to the growing fetus [1,2]. Decreased insulin sensitivity leads to beta-cell proliferation and a larger volume of individual beta-cells, returning to non-pregnant levels after parturition. Hyperglycemia may develop due to insufficient beta-cell proliferation [3,4,5,6]. However, the mechanism behind insulin resistance is multifactorial. Women with a body mass index (BMI) ≥ 25 kg/m^2^ have excess adipose tissue which is known to reduce insulin sensitivity and may explain the correlation with adverse outcomes of pregnancy for both mother and child [7]. The worldwide prevalence of overweight and obesity during pregnancy is rapidly increasing, with a prevalence of nearly 54 million worldwide [8]. In the Netherlands, the prevalence of overweight women age 30–40 years was 39% in 2017 and around 44% in 2019 [9], suggesting that many women are already overweight when becoming pregnant. The physiological insulin resistance during pregnancy in combination with overweight or obesity could increase the risk for hyperglycemia. The hyperglycemia and pregnancy outcomes (HAPO) study has shown that already small increases in maternal glucose levels show a linear relationship with adverse outcomes [10].

Insulin resistance during pregnancy increases the risk of pregnancy complications such as pregnancy induced hypertension, pre-eclampsia, caesarian section and gestational diabetes mellitus (GDM), and excessive pregnancy weight gain during pregnancy and weight retention postpartum, while longer term it also increases the risk of cardiovascular diseases and diabetes mellitus (DM) type 2 and other non-communicable diseases. Adverse outcomes of pregnancy hyperglycemia in infants could be macrosomia and large for gestational age (LGA), as well as neonatal complications on the short term and a higher risk on childhood obesity and non-communicable diseases longer term [7].

Maintaining euglycemia despite the developing insulin resistance over the course of pregnancy is critical. Deviations in maternal glucose metabolism could potentially already be detected early in pregnancy [7,10,11]. Yet, potential hyperglycemia is currently standardly examined at the end of the second trimester by an oral glucose tolerance test (OGTT), and in the Netherlands only in an at-risk population defined as BMI ≥ 30 kg/m^2^ [12]. This test, taking baseline and 2-h blood glucose levels as a result, is not suitable to detect mild hyperglycemia in early pregnancy and shows considerable within subject variability [13]. Markers of insulin sensitivity and related metabolic adaptations, for instance in lipid metabolism, may be a more straightforward measure in which deviations could potentially be detected earlier [14]. An integration of postprandial responses of glucose and insulin combined with lipid markers following a challenge test would provide clearer insights in maternal metabolic function. To this end, a meal tolerance test (MTT), which contains a well-defined balanced macro- and micronutrient composition may be much more sensitive than the standard OGTT. Assessing glucose homeostasis based on glucose concentrations only is not reliable as there are numerous perturbations where glucose production and its utilization increases or decreases to the same extent without any changes in blood concentrations [15]. For the understanding of the physiology and pathophysiology of glucose uptake and metabolism during pregnancy, glucose tracers could be used [16].

The design of the Pregnancy Outcomes and Maternal Insulin Sensitivity (PROMIS) study will allow collection of detailed insights in insulin sensitivity and glucose metabolism during (early) pregnancy in women with overweight and obesity and assess its possible relation to growth and body composition during of the offspring during the fetal and early infancy period. We hypothesize that using an MTT, disturbed insulin sensitivity in overweight pregnant women could be identified much earlier in pregnancy. Early detection could help to better predict the risk of possible adverse outcomes short term and reduce the risk of non-communicable diseases on the longer term.

## 2. Materials and Methods

### 2.1. Overall Study Design

The PROMIS study is a multi-center exploratory prospective cohort-based study in the two Northern provinces of the Netherlands (Groningen and Friesland). The study has been approved by the medical ethical committee University Medical Centre Groningen prior to the start of the study (NL68845.042.19) and is registered at ClinicalTrials.gov (NCT04315545). Healthy pregnant women with a BMI ≥ 25 kg/m^2^ are eligible for participation. Weight and height of women as well as fasting glucose levels will be confirmed via measurements by study staff at inclusion. Study duration will be from 12 weeks of gestation up to 6 months after giving birth with a total of nine contact moments when data will be collected. The MTT will be performed between weeks 12 and 16 and repeated between weeks 24 and 28 of gestation and three months postpartum. A standard OGTT will be performed after the second MTT, also between weeks 24 and 28 of gestation. In addition, maternal anthropometrics will be assessed at the time of the MTT challenges. Additional sonographies will be made during gestation at week 28 and 32 to assess fetal growth. Neonatal and infant anthropometrics will be obtained after birth and at 1, 2, 3, 4, and 6 months of age from the records of the baby clinic visits. Body composition measurements using an air displacement plethysmography (Pea Pod, COSMED, Rome, Italy) [17] will be performed at 1, 3, and 6 months of age. Questionnaires about nutrition and lifestyle will also be collected at several time points throughout the study. A study overview is depicted in Figure 1.

### 2.2. In-and Exclusion Criteria

Inclusion criteria are singleton pregnancy, BMI ≥ 25 kg/m^2^, fasted (≤7.0 mmol/L) or random (≤11.0 mmol/L) blood glucose level, aged ≥ 18 years, and written informed consent. Women will be excluded from the study if they have pre-existing DM type 1 or 2, serious health complications (hypertension, hyperlipidemia, asthma, haemochromatosis) or medication use that influences the glucose metabolism or fetal growth (e.g., chronic use of corticosteroids). In addition, women who are participating in any other studies involving investigation of medication or nutritional products, have a severe illness, psychological dysfunctions, used antibiotics two weeks prior to participation, have HIV or hepatitis, expected to not comply to the study protocol (e.g., fear of needles), known allergies or intolerances for one or more nutritional ingredient used in the MTT will also be excluded.

### 2.3. Recruitment

Women will be recruited in the provinces of Groningen and Friesland from a diverse range of socio-economic and ethnic backgrounds. Hospitals, midwife practices, ultrasound practices, and family doctor’s practices are involved in recruitment. To optimize recruitment, flyers were made, and social media accounts set up. Information sheets explaining aspects of the study, as well as a short video explaining the study set up will be provided to eligible participants. Women will have the opportunity to consider participation and to ask questions to the researchers and can terminate participation at any time within the study without affecting their usual care. The privacy of the participant will be ensured via protected data collection and processing by the researchers and will be saved anonymously. Written informed consent will be obtained prior to the first baseline measurement. Women are requested to give consent for (1) participation in the study, including taking of blood samples, collecting associated measurements, and extraction of relevant data from medical records; (2) storage of blood in a bio bank and (3) participation of their child in this study including gathering protocol specified measurements.

### 2.4. Meal Tolerance and Oral Glucose Tolerance Testing

To conduct the MTT, a non-investigational product, Nutridrink Compact, will be used. Nutridrink Compact is a liquid product for medical use to treat malnutrition. It provides energy and nutrients in a relatively compact volume and contains a suitable combination and levels of macro- and micronutrients, largely comparable to the energy and nutrient composition of a standard breakfast. The product volume will be standardized to 50 gr carbohydrates and contains 24 g of protein and 9 g of fat, equivalent to 404 kcal in a total volume of 169 mL. The study participants are asked not to eat or drink (except for water) anything from 11 pm before the test day to arrive in the fasted state. For regular OGTT testing, a standard 75 gr glucose solution of 200 mL will be used. Participants will be diagnosed with GDM when plasma blood glucose is ≥7.0 mmol/L in fasted state or ≥7.8 mmol/L 2 h after the OGTT. These guidelines are according to the Dutch Association of Obstetrics and Gynecology (NVOG) [18]. In a subgroup of participants (estimate *n* = 20) we will add 2% of the total amount of carbohydrates in the MTT as glucose tracers. In total, 1.25 g of D-(6,6-D2, 99%)-glucose, used as the tracer, will be added to the MTT, as well as to the OGTT, selected to ensure participant and experimental safety [19]. The solution will be prepared by the research pharmacy in special units that require only the addition of the required MTT or OGTT volume. In both the MTT and the OGTT, the participant should ingest the drink within 5 min.

### 2.5. Sample Size Calculation

The primary outcome of the PROMIS study is to detect changes in blood glucose- and insulin levels following an MTT, containing 50 g of carbohydrates next to other nutrients. The null hypothesis is that early pregnancy assessment of maternal glucose-insulin metabolism with an MTT will identify more mothers at risk for pregnancy complications compared to a standard OGTT conducted at the end of the second trimester. The current prevalence of GDM, a state of reduced insulin sensitivity leading to hyperglycemia based on standard OGTT testing, is estimated to be 3–5% in the Netherlands, as diagnosed between 24–28 weeks of gestation. Given the reported linear relationship between blood glucose and adverse outcomes [10] and higher rates of insulin resistance with increased BMI [20], more heterogeneity, disturbances in blood glucose, and insulin values are expected in our study population than the indicated 3–5% [21] for the total population of pregnant women in the Netherlands. In addition, we anticipate that the first abnormal blood values, in particular signs of insulin resistance might already be detected in early gestation using this more physiological testing paradigm. To be able to reject the null hypothesis, we estimate that a total of 130 pregnant women are needed to participate in the PROMIS study, allowing for a dropout rate of 20%. This is based on an expected sensitivity of 74% and a minimal lower confidence limit of 0.62 in OGTT’s containing 50 g of carbohydrates, given before 32 weeks of gestation for observational studies [22,23]. Given the exploratory nature of the study, the sample size will be re-estimated after the first OGTT has been completed in 25 subjects based on available sensitivity and specificity data.

### 2.6. Interim Analysis

After including a total of 5 participants that completed the first MTT, a first interim analysis will be performed to see if the added amount of glucose tracer is indeed sufficient to trace glucose fate following the MTT. If the amount of glucose tracer is insufficient for analysis, the size of the subgroup and the percentage of added glucose tracer can be adjusted. The first interim will also be used to assess possible unexpected adverse events. A second interim analysis will be done after the completion of the OGTT in the first 25 subjects to recalculate the sample size by sensitivity and specificity after confirmation of GDM incidence based on blood glucose levels. It is expected that the final number for inclusion will be lower than the calculated 130 participants given the detailed data collection to assess glucose-insulin metabolism.

### 2.7. Data Collection

An overview of all the parameters measured during pregnancy in the mother and fetus (Table 1) and postpartum in mother and child (Table 2) is provided below.

#### 2.7.1. Maternal Measurements

Basal (*t* = 0), as well as postprandial blood samples, will be collected at the following time points; 10, 20, 30, 45, 60, 90, and 120 min after consumption of the test meal. In fasted state, the following blood parameters will be measured; glucose, labeled glucose, insulin, hbA1c, triglycerides, free fatty acids, total cholesterol, HDL-cholesterol, HPL, c-peptide, and cortisol. Postprandial blood measures include glucose, glucose tracers, and insulin. Several blood parameters will be directly analyzed by the routine laboratory available in the hospitals, while other blood collection tubes will be directly centrifuged for 10 min at 1300× *g*, divided into aliquots varying from 500 uL to 2 mL and stored in a −80 °C freezer for later analysis. Glucose, total cholesterol, and HDL-cholesterol will be analyzed by photometrics, insulin, and cortisol will be analyzed by luminescent-immuno-assay, HbA1c will be analyzed by liquid chromatography, leptin, free fatty acids, and HPL will be analyzed by the ELISA method and C-peptide will be analyzed by Immuno-assay. To measure the labeled glucose, blood spots of 75–80 uL will be taken to onto 903 protein saver cards (Whatman [24]). Fractional contributions of glucose tracer in blood will be measured by gas chromatography mass spectrometry (GCMS) according to van Dijk et al. (2003) [16]. The collected data will be used in the minimal model adapted from Cobelli et al. (2009) [15]. This mathematical model allows estimation of endogenous glucose production and clearance rates at fasting as well as postprandial states. The fractional absorption rate, delay time, and apparent volumes of distribution of the administered bolus can be estimated. For the utilization parameters, peripheral insulin sensitivity, and glucose effectiveness will be estimated, as well as hepatic insulin sensitivities.

From the postprandial time points, the area under the curve (AUC), peak, insulin sensitivity index and deltas can be calculated. The calculation of the homeostatic model assessment for insulin resistance (HOMA-IR) will be made based on the fasting plasma glucose (FPG) and insulin (FPI) values (FPG × FPI/22.5). The quantitative insulin sensitivity check index (QUICKI) is based on fasting blood samples and is calculated using the following formula QUICKI = 1/(log(I0) + log(G0)), in which I0 is the fasting insulin, where G0 is the fasting glucose value.

Maternal anthropometric measures that will be collected include height, weight, pre-pregnancy weight, (self-reported) pre-pregnancy weight, gestational weight gain, circumferences, and maternal fat mass. BMI is calculated as weight divided by square heights. Measured circumferences include neck, waist, thigh, hip, wrist, and upper arm. The hip–waist ratio will also be calculated. Maternal fat mass will be estimated using skinfold thickness measurements by Harpenden Skinfold Caliper (Baty international [25]. Skinfold thickness measurements will be done at one side of the body for three times. The average of those three skinfolds will be used. The skinfolds will be obtained twice during pregnancy at the time of the challenge tests, between 12–16 weeks and between 24–28 weeks, and postpartum after 3 and 6 months. Skinfolds to assess adiposity have been validated in pregnant women and will be measured at 4 locations; (1) triceps, (2) biceps, (3) subscapular skinfold, and (4) suprailiac skinfold [26]. Skinfolds will be calculated by the equations of Durnin and Womersley [27].

#### 2.7.2. Fetal Measurements

Fetal growth characteristics including crown and rump length, biparietal diameter (BPD), head circumference (HC), abdominal circumference (AC), Femur Length (FL), and occipito-frontal diameter (OFD) will be obtained through sonography by trained professionals in weeks 28 and 32 of gestation. The INTERGROWTH-21 [28] provides additional data to the standard growth standards of the world health organization (WHO) [29] and will be used for the estimation fetal growth by filling in the calculator of gestational age and estimated weight of fetus. The ultrasounds will be performed in certified ultrasound (midwifery) centers by their own certified staff.

#### 2.7.3. Neonatal and Infant Measurements

Neonatal and infant data collection consists out of birth weight taken from antenatal records, anthropometrics (height, weight), circumferences (head, upper arm, wrist, thigh, neck, waste), and skinfolds (biceps, triceps, supra-illiac, thigh) at 1, 2, 3, 4, and 6 months of age. Body composition measurements will be performed using air displacement plethysmography at 1, 3, and 6 months of age. Data output will include calculated estimates of fat mass and fat free mass, as well as body volume and body density.

#### 2.7.4. Diet and Behavioral Data

Specific validated questionnaires will be given to the participants to gain insight in different diet and behavioral data at different times throughout the pregnancy and postpartum period. A food frequency questionnaire (FFQ) which consist out of items related to different food groups, will be used to measure food intake, and calculate energy and macronutrient intake in the mother over the last month [30]. The adult eating behavior questionnaire (AEBQ) will be given to study eating habits, it consists out of 34 items with five answer options ranging from strongly disagree to strongly agree [31]. The EuroQol-5D (EQ-5D) which consist of five dimensions regarding mobility, self-care, daily activities, pain, and mood to measure quality of life will be measured [32]. The pregnancy physical activity questionnaire (PPAQ) will be used to measure overall activity duration and intensity during pregnancy. It consists out of 34 items with six answering options [33]. After pregnancy, the international physical activity questionnaire (IPAQ) will be used, which consists out of 7 items where participants can fill in a certain number of hours and minutes [34]. In addition, we will use a short questionnaire to collect data on the consumption of the MTT and OGTT. This ‘sensory questionnaire’ is used to determine which challenge test is preferred. It consists out of 34 items ranging in different variables. For the infant we use a questionnaire about the type of feeding the baby is receiving (e.g., breastfed vs. formula fed), as well as the baby eating behavior questionnaire (BEBQ). This questionnaire consists out of 18 items with five answer options ranging from never to always [35].

#### 2.7.5. Other Outcomes

Social demographics recorded include marital status, household composition, occupation, ethnicity, financial household income, level of education, and paternal information. The questionnaire used includes family history of DM, previous GDM diagnosis, birth weight of previous pregnancy (if any), more detailed questions about previous child or children born, polycystic ovary syndrome (PCOS), medication use, pregnancy eclampsia, and pre-pregnancy hypertension. In addition, obstetrical and perinatal complications during and after pregnancy will be registered, such as spontaneous or induced abortion, intrauterine death, preeclampsia, pre-term delivery (<37 weeks), need for labor induction, assisted vaginal delivery, shoulder dystocia, caesarian section. Finally, adverse events related to the neonate/infant such as macrosomia, large for gestational age, small for gestational age, hypoglycemia, respiratory distress, neonatal hospitalization, and other complications (if any) will be recorded.

### 2.8. Statistical Analysis

Data will be expressed as mean ± Standard Deviation (SD) or ranges for continuous variables. Categorical variables will be presented as percentages. All statistical comparisons are considered significant at *p* ≤ 0.05 (two-sided). Descriptive statistics will present averages and differences at baseline between medium (BMI ≥ 25 kg/m^2^), and high risk (BMI ≥ 30 kg/m^2^) groups in our study population. The significance of the mean differences will be tested using analysis of variance (ANOVA). Differences between categorical variables will be tested using the k^2^ test when appropriate. Associations will be tested and corrected for selected confounders and multiple testing where appropriate. Mixed models will be used to measure postprandial responses. A statistical analysis plan will be made for each interim analysis as well as for more detailed analysis of all collected parameters before data base closure.

### 2.9. Database

All participant data will be entered into Research electronic data capture (REDCap, Vanderbilt University) [24]. Operatory entry is password protected. Pages are in English and consist of drop-down menu’s, pick lists, and text boxes. No personal study subject identifiers can be found in the database, data will be entered under study ID number, wherein the first number identifies the study location.

## 3. Discussion

The PROMIS study is an exploratory cohort study wherein the primary outcome is the association between fasting and postprandial glucose and insulin, using the AUC in relation to fetal growth and birth weight. Secondary parameters including plasma lipids and (placental) hormones, growth and adiposity will be analyzed in relation to glucose-insulin outcomes in different stages during and after pregnancy. Exploratory study parameters to assess glucose- and insulin metabolism include analysis of glucose tracers and modeling glucose disappearance rates, as well as lifestyle parameters and the food-and behavior questionnaires.

The PROMIS study investigates early insulin resistance during pregnancy focusing on healthy women with an increased risk of adverse outcomes. The WHO guidelines already consider women with a BMI ≥ 25 kg/m^2^ to have an increased risk of developing GDM [36]. In the Netherlands, however, only overweight defined as BMI equal or above 30 is considered as a risk factor for GDM [18], excluding specifically women with a BMI between 25 and 30 from screening for GDM during the second trimester of pregnancy. This is one of the first studies to focus on assessment of metabolic parameters in early pregnancy as possible predictors for adverse outcomes in both mother and child.

Current guidelines for the diagnosis of GDM focus on the inability of the maternal metabolism to cope with changes in insulin sensitivity and insulin demand. From the HAPO study results it is clear that the linear relationship between blood glucose levels in the second trimester of pregnancy and different short term (birth weight, C-section), as well as long term (child growth, insulin sensitivity) outcomes may advocate a different approach if we aim to prevent some of these pregnancy complications in a following study [10]. The focus on glucose rather than on the underlying adaptations in insulin production that are needed to compensate for the changes in insulin sensitivity, make it impossible to diagnose those at risk for adverse outcomes earlier in pregnancy, unless the metabolic deviations are severe. As insulin sensitivity also strongly influences the nutrient flow of the placenta to the fetus, diagnosis around the second trimester may simply be too late to prevent adverse consequences for the growing fetus. In most countries, GDM is diagnosed between week 24–28 of gestation only in women with agreed risk factors, such as obesity (BMI ≥ 30 kg/m^2^), previous GDM, LGA, small for gestational age (SGA), first family relative with diabetes, ethnic background, or PCOS [18]. The standardly used OGTT containing 75 gr of glucose dissolved in water has been subject of much debate [37,38]. By using an MTT in the present study, which contains a balanced macronutrient composition of proteins and fats besides carbohydrates, we will challenge the glucose metabolism, as well as the insulin production and the response of the lipid metabolism. In this way, we hope to collect more accurate and reliable insights in metabolic function already much earlier in pregnancy and generate more reliable data using a much lower and physiological metabolic load challenge test.

The design of the PROMIS study has some potential weaknesses. First, we intend to recruit women with a risk factor for metabolic deviations in a cohort like approach rather than using a randomized trial design directly comparing groups from different BMI classes. Although the latter may use accepted standard definitions, BMI may not adequately represent body adiposity, the major driver of insulin sensitivity. The choice for focus on more detailed assessment in women with a clear risk for increased adiposity may reduce the total number of women we need to include for adequate analysis of the results. The current power estimate, however, which is based on glucose data only, may not allow for powered analysis for many of the secondary outcomes. Secondly, the detailed and repeated measurements may form a burden for participants increasing the risk of drop out. We hope to overcome this hurdle by offering specific assessments with attention for health of mother and her infant next to the routine care these women receive during and after pregnancy. The detailed assessment of glucose, as well as insulin values using multiple time points during the postprandial phase, as well as the addition of stable labelled glucose to assess other glucose parameters, i.e., glucose production and disappearance rates, is a clear strength of this study. The repeated assessment will allow for longitudinal assessment of pregnancy related changes in glucose-insulin metabolism, as well as within subject comparison of results between MTT and OGTT, where the MTT provides a more physiological challenge compared to the standard OGTT. Finally, the detailed measurements of fetal as well as early postnatal growth and body composition in the infants may generate insights in early growth trajectories in relation to maternal glycemia and other metabolic markers. These data may help to develop markers for early detection of metabolic deviations and improve the diagnostic toolbox, as well as inform the development of new interventions, although these would require in depth validation in new clinical studies.

## 4. Conclusions

The design of the PROMIS study will generate insight in early pregnancy metabolic changes in overweight and obese women that may predict later pregnancy and postpartum maternal outcomes and possible associations with offspring growth and body composition outcomes. These data may provide a starting point for future design of focused intervention studies to prevent and manage maternal as well as fetal and infant complications.

## Figures and Tables

**Figure 1 jcm-10-00976-f001:**
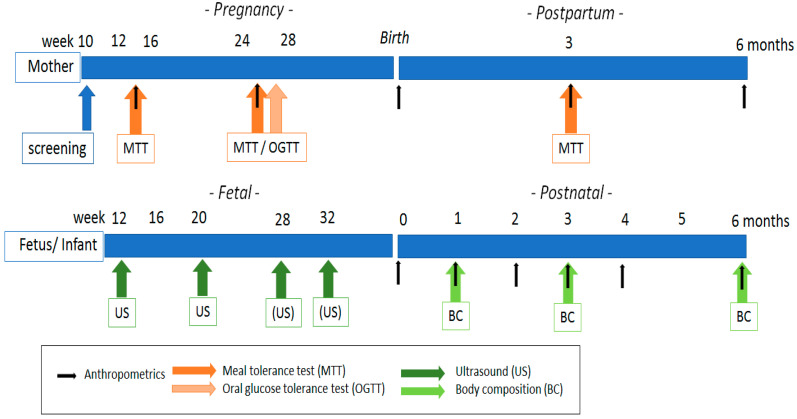
Schematic representation of the prospective cohort indicating readouts and proposed timing of measurement between early pregnancy up to 6 months postpartum in both mother (upper panel) and offspring (lower panel).

**Table 1 jcm-10-00976-t001:** Timing of data collection during pregnancy up to 2 weeks postpartum in mother and fetus-infant.

	Screening	Week 12–16	Week 20	Week 24–26	Week 28	Week 32	Week 36	Birth
Visit number	Visit 1	Visit 2	Routine ^1^	Visits 3 and 4	Visit 5	Visit 6	Phone call	Routine ^2^
Visit window	≤12 weeks	±1 week	±2 weeks	±1 week	±1 week	±1 week	±1 week	+2 weeks
Informed consent	X							
Inclusion/exclusion criteria	X							
Baseline and demographics		X						
Maternal anthropometrics	X	X		X			X	
Fasting or random blood glucose	X							
Blood parameters (for MTT)		X		X				
Blood parameters (for OGTT)				X				
Maternal skinfolds and fundal height		X		X				
Fetal sonography			X		X	X		
FFQ		X		X				
PPAQ		X		X		X		
Sensory questionnaire				X				
EQ-5D		X		X		X		
AEBQ		X		X				
Birth outcomes								X
Neonatal adiposity and anthropometrics								X
Intake drugs, alcohol, smoking	X			X				
Maternal complications	X	X		X			X	X
Neonatal complications								X

^1^ Routine care includes standard sonography measure, ^2^ Routine postpartum care is performed at home or at the neonatal clinic, OGTT: oral glucose tolerance test, MTT: mixed meal tolerance test, PPAQ: pregnancy physical activity questionnaire, EQ-5D: euroQol-5D, AEBQ: adult eating behavior questionnaire, FFQ: food frequency questionnaire, method to assess dietary intake in mothers, IPAQ: international physical activity questionnaire.

**Table 2 jcm-10-00976-t002:** Timing of data collection up to 6 months postpartum.

	1 Month Postpartum	2 Months Postpartum	3 Months Postpartum	4 Months Postpartum	6 Months Postpartum
Visit number	Visit 7	Routine ^1^	Visit 8	Routine ^1^	Visit 9
Visit window (days)	±1 week	±3 days	±1 week	±3 days	±1 week
Maternal anthropometrics			X		X
Blood parameters MTT			X		
Skinfolds			X		X
Food Frequency Questionnaire	X		X		X
IPAQ			X		X
EQ-D5	X		X		X
AEBQ	X		X		X
BEBQ	X		X		X
Infant adiposity and anthropometrics	X	X	X	X	X
Peapod/skinfolds	X		X		X
Maternal complications	X	X	X	X	X

^1^ Routine care postpartum is performed at home or the neonatal clinic, MTT: mixed meal tolerance test, PPAQ: pregnancy physical activity questionnaire, EQ-5D: euroQol-5D, AEBQ: adult eating behavior questionnaire, 24-h recall: method to assess dietary intake in mothers, IPAQ: international physical activity questionnaire, BEBQ: baby eating behavior questionnaire.

## Data Availability

No new data were created or analyzed in this study. Data sharing is not applicable to this article.

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
