# Peer review of "Pregnancy Outcomes and Maternal Insulin Sensitivity: Design and Rationale of a Multi-Center Longitudinal Study in Mother and Offspring (PROMIS)"

_jcm, 2021, doi:10.3390/jcm10050976_

Round 1

Reviewer 1 Report

This is a well written and clear manuscript detailing the design and rationale of the PROMIS study. The study is described well and is an important topic with the rate of overweight/obese women and therefore GDM increasing. The overall rationale of the study, to investigate early metabolic changes in these women could lead to improvement in diagnostics. The main changes that have been recommended before this manuscript is published are that you need to make the reader aware in the abstract and in the introduction that this is paper is describing the design of the study and will not go in to any results. Consequently, having a results section is misleading and I would suggest that the information in this section be moved into the Methods section, as this manuscript is just detailing methods without results.

Minor revisions:

  1. Abstract: it is not clear from the abstract that this manuscript focuses on the design and rationale of the study, rather than the results of the study. This needs to be stated more clearly in the abstract (and introduction).
  2. Line 90-93: This sentence is a little jumbled and could do with being restructured so that the hypothesis is clearer.
  3. Method: Recruitment: when will the recruitment begin? Has it already begun? If so, how many women have been consented so far?
  4. Table 1 – needs to be reformatted, hard to read
  5. It would be interesting in the discussion to add some details regarding the strengths and weaknesses/limitations of the study, ie. where you may anticipate problems in the study design and how you will overcome these potential issues

Major revisions:

  1. As this manuscript is detailing the design and rationale of the PROMIS study it is misleading to have a results section as you do not yet have results for the study. Therefore, the Results title should be removed and there should be one larger Materials and Methods section with all the details of the study, including what is in the current results section

Author Response

We thank the reviewer for his/her positive feedback and valuable suggestions to improve the manuscript.

Major revisions: As this manuscript is detailing the design and rationale of the PROMIS study it is misleading to have a results section as you do not yet have results for the study. Therefore, the Results title should be removed and there should be one larger Materials and Methods section with all the details of the study, including what is in the current results section

We agree with the reviewer that since the paper describes the rationale and design of the study and does not report actual study results, the use of a results section in the manuscript may confuse the reader. We have therefore adapted this and included the information now under the materials and methods section, adjusting also the numbering of the different headings and subheadings using track changes. We decided that the last section on study outcomes, describing the primary, secondary and exploratory aims of the study had a better fit with the discussion section of the manuscript and moved that part to become the opening paragraph of the discussion in the revised version of the paper.

Minor comments:

  1. Abstract: it is not clear from the abstract that this manuscript focuses on the design and rationale of the study, rather than the results of the study. This needs to be stated more clearly in the abstract (and introduction).

We have tried to formulate the sentences in such a way that it is clear that the description concerns a detailed explanation of the rationale and study design and not reporting of any results. To prevent any doubt, however, we have now adapted the abstract and the introduction as follows:

Abstract, line 16-18:

The aim of the PROMIS study is to obtain detailed insights in insulin sensitivity and glucose metabolism during pregnancy and its relation to pregnancy outcomes including early infancy growth. The PROMIS study is an exploratory cohort study recruiting healthy pregnant women

We changed the current sentence into:

“The present paper describes the rationale and design of the PROMIS study, an exploratory cohort study to obtain detailed insights in insulin sensitivity and glucose metabolism during pregnancy and its relation to pregnancy outcomes including early infancy growth. We aim to recruit healthy pregnant women….”

Abstract, line:  32-38

The PROMIS study will provide detailed insights in the metabolic changes in the mother and their possible association with fetal and postnatal infant growth and body composition. The data from this cohort of healthy pregnant women with overweight as a risk factor for the development of GDM may provide new insights to detect metabolic deviations already in early pregnancy and help to develop interventions to improve the management of maternal as well as fetal/infant complications from already early on with the aim to prevent adverse outcomes for mother and child.

We changed the current paragraph into:

The design of the PROMIS study will allow for detailed insights in the metabolic changes in the mother and their possible association with fetal and postnatal infant growth and body composition. We anticipate that the data from this cohort of women with an elevated risk for the development of GDM may provide new insights to detect metabolic deviations already in early pregnancy. These data could inspire the development of new interventions that may improve the management of maternal as well as fetal/infant complications from already early on in pregnancy with the aim to prevent adverse outcomes for mother and child.”

Introduction, line 87-92

The aim of the PROMIS study is to obtain detailed insights in insulin sensitivity and glucose metabolism during (early) pregnancy in women with overweight and obesity in relation to growth and body composition during fetal and early infancy period in the offspring.

We changed this paragraph into: “The design of the PROMIS study will allow collection of detailed insights in insulin sensitivity and glucose metabolism during (early) pregnancy in women with overweight and obesity and assess its possible relation to growth and body composition development of the offspring during the fetal and early infancy period.”

  1. Line 90-93: This sentence is a little jumbled and could do with being restructured so that the hypothesis is clearer.

We hypothesize that using an MTT, disturbed insulin sensitivity in overweight pregnant women could be identified much earlier in, making it possible to better predict the risk of possible adverse pregnancy outcomes short term and non-communicable diseases on the longer term.

We changed the current hypothesis description into:  

“We hypothesize that using an MTT, disturbed insulin sensitivity in overweight pregnant women could be identified much earlier in pregnancy. Early detection could help to better predict the risk of possible adverse outcomes short term and reduce the risk of non-communicable diseases on the longer term.”

  1. Method: Recruitment: when will the recruitment begin? Has it already begun? If so, how many women have been consented so far?

Initial recruitment of participants into the study started in January 2020, but only 2 subjects were included when the national lock down as a result of the COVID situation in the Netherlands complicated measurements as well as further inclusion. Due to the evolving COVID situation in the Netherlands over the past year, inclusion of participants was off and on possible during short time windows. So far only 12 subjects have been included in the study. We hope the situation will slowly normalize and that we can (re)start recruitment during the course of 2021.

  1. Table 1 – needs to be reformatted, hard to read

We agree that the readability of the tables could be approved and have adapted the design and line distance in both table 1 and 2.

  1. It would be interesting in the discussion to add some details regarding the strengths and weaknesses/limitations of the study, ie. where you may anticipate problems in the study design and how you will overcome these potential issues

We thank the reviewer for this suggestion and agree that a short discussion on strengths and weaknesses may add value to the paper. We have now added the following paragraph to the discussion section, just prior to the conclusions section:

“The design of the PROMIS study has some potential weaknesses. First, we intend to recruit women with a risk factor for metabolic deviations in a cohort like approach rather than using a randomized trial design directly comparing groups from different BMI classes. Although the latter may use accepted standard definitions, BMI may not adequately represent body adiposity, the major driver of insulin sensitivity. The choice for focus on more detailed assessment in women with a clear risk for increased adiposity may reduce the total number of women we need to include for adequate analysis of the results. The current power estimate, however, which is based on glucose data only, may not allow for powered analysis for many of the secondary outcomes. Secondly, the detailed and repeated measurements may form a burden for participants increasing the risk of drop out. We hope to overcome this hurdle by offering specific assessments with attention for health of mother and her infant next to the routine care these women receive during and after pregnancy.The detailed assessment of glucose as well as insulin values using multiple time points during the postprandial phase as well as the addition of stable labelled glucose to assess other glucose parameters, i.e. glucose production and disappearance rates, is a clear strength of this study. The repeated assessment will allow for longitudinal assessment of pregnancy related changes in glucose-insulin metabolism as well as within subject comparison of results between MTT and OGTT, where the MTT provides a more physiological challenge compared to the standard OGTT. Finally, the detailed measurements of fetal as well as early postnatal growth and body composition in the infants may generate insights in early growth trajectories in relation to maternal glycemia and other metabolic markers. These data may help to develop markers for early detection of metabolic deviations and improve the diagnostic toolbox, as well as inform the development of new interventions, although these would require in depth validation in new clinical studies.”

Reviewer 2 Report

This is a well written, well exposed paper of a study design. 

I only have minor details to mention:

Abstract:

“to exclude previously undiagnosed types of diabetes”, I would say: to exclude pregestational diabetes

Last sentence of the abstract in the aims of the study:

“The data from this cohort…may provide detailed insights … and help to develop interventions to improve…”    I think this last aim will not be accomplished from this study- It is not precise- it is better to state that it may help identify risk factors and eventually design intervention studies in the future aimed to improve.. but from identification of risk and prevention there is a large journey. This is very well explained later on the paper: Line 359: Conclusion

Line 208:   “in fasted state”  should be: In the fasted state

3.8 Outcome measures. I found this should be a little bit more extended. It has been stated in other parts of the paper but it looks like in this specific part, it has been too much summarized (it is better to mention postprandial glucose and AUCs after MTT and OGTT

-Have you considered extending this study to longer term?- 2 yr- 6 years of age of the offspring? it would be very interesting

Author Response

We thank the reviewer for his positive feedback suggestions for improvement and have made the following adaptations to the manuscript:

  • Abstract: “to exclude previously undiagnosed types of diabetes”, has been changesd into ‘to exclude pregestational diabetes’

  • Last sentence of the abstract in the aims of the study:

“The data from this cohort…may provide detailed insights … and help to develop interventions to improve…”    I think this last aim will not be accomplished from this study- It is not precise- it is better to state that it may help identify risk factors and eventually design intervention studies in the future aimed to improve.. but from identification of risk and prevention there is a large journey. This is very well explained later on the paper: Line 359: Conclusion

We appreciate the suggestion of the reviewer and have adapted the concluding sentence of the abstract also based on other feedback received as follows: “These data could inspire the development of new interventions that may improve the management of maternal as well as fetal/infant complications from already early on in pregnancy with the aim to prevent adverse outcomes for mother and child.”

Line 208:   “in fasted state”  should be: In the fasted state – adapted in the current version

3.8 Outcome measures. I found this should be a little bit more extended. It has been stated in other parts of the paper but it looks like in this specific part, it has been too much summarized (it is better to mention postprandial glucose and AUCs after MTT and OGTT

We have made a number of adaptations to this part of the paper also based on the suggestions of other reviewers. We prefer, however, to keep the current order of sub-headings, as the data analysis details are now grouped under maternal and fetal/infant outcomes and thus have a better fit with the timing and outcome summary provided in table 1 and 2 respectively.

-Have you considered extending this study to longer term?- 2 yr- 6 years of age of the offspring? it would be very interesting

We agree with the reviewer that it would be very interesting to follow up the infants until later in childhood. We indeed are discussing to ask the participants if they are willing to participate in a follow up study (rather than extending the main study). However, actual conduct of such a long term outcomes study will only happen if we can secure the funding needed to conduct follow up assessments.

Reviewer 3 Report

The authors provide a review of the proposed PROMIS trial. They aim to complete an exploratory cohort study of pregnant women with BMI>25 before 12 weeks of gestation, in which they will examine glucose values, lipid profiles, anthropometric measurements, and neonatal data. With their results, they hope to show deviations in metabolic function that exist in early pregnancy and help to develop interventions to improve complications as a result of this.

Introduction: The intro is well written, and provides sufficient citation of pre-existing literature regarding this topic. Some citation regarding why lipid metabolism is a more straightforward measure should be included as this is the basis for proposing use of the MTT.

Methods:  Very clearly written with good use of additional tables and figures to show time points of interventions. It may be worth explaining why a second MTT is done in addition to OGTT.

Results: This section is well done with good explanations of anticipated outcome measures.

Discussion: Highlights the implications of their potential findings and how it may impact clinical care.

There are a few places that require rewording for clarity of sentences (lines 54-55, lines 62-63, line 88, line 332).

Author Response

We thank the reviewer for his/her positive and constructive feedback on the paper. We appreciate the suggestions to review the indicated sentences that may need rewording to improve clarity and have made the following adaptations to the paper:

lines 54-55: The worldwide prevalence of overweight and obesity during pregnancy is rapidly increasing, with a prevalence of nearly 54 million worldwide [8]. The prevalence of overweight women between 30-40 years in the Netherlands, for instance, was 39% in 2017 [9].

We changed into: “The worldwide prevalence of overweight and obesity during pregnancy is rapidly increasing, with a prevalence of nearly 54 million worldwide [8]. In the Netherlands, the prevalence of overweight in women age 30-40 years was 39% in 2017 and around 44% in 2019 [9], suggesting that many women are already overweight when becoming pregnant.

Lines 62-63: Adverse maternal outcomes due to insulin resistance could be pregnancy induced hypertension, pre-eclampsia, caesarian section and gestational diabetes mellitus (GDM) on the short term and increased risk of weight retention and non-communicable diseases like cardiovascular diseases and diabetes mellitus (DM) type 2 on the longer term.

We changed into: Insulin resistance during pregnancy increases the risk for pregnancy complications such as pregnancy induced hypertension, pre-eclampsia, caesarian section and gestational diabetes mellitus (GDM) and excessive weight gain during pregnancy and weight retention postpartum, while on the long term it also increases the risk of cardiovascular disease, diabetes mellitus (DM) type 2 and other non-communicable diseases.”

 line 88: The aim of the PROMIS study is to obtain detailed insights in insulin sensitivity and glucose metabolism during (early) pregnancy in women with overweight and obesity in relation to growth and body composition during fetal and early infancy period in the offspring.

We changed into: “The design of the PROMIS study will allow collection of detailed insights in insulin sensitivity and glucose metabolism during (early) pregnancy in women with overweight and obesity and assess its possible relation to growth and body composition development of the offspring during the fetal and early infancy period.”

line 332: Overweight is, however, not considered as a risk factor for GDM in the Netherlands at the moment [17], excluding especially women with a BMI between 25 and 30 from screening for GDM during the second trimester of pregnancy.

We changed into: “In the Netherlands, however, only overweight BMI>30 is considered as a risk factor for GDM [17], excluding specifically women with a BMI between 25 and 30 from screening for GDM during the second trimester”.

Finally, we have included an additional reference [numbered now as 14] to support the notion that lipid metabolism may be a more straightforward measure as suggested (see Introduction, line 93 revised version)

  1. Benhalima K, Van Crombrugge P, Moyson C, Verhaeghe J, Vandeginste S, Verlaenen H, Vercammen C, Maes T, Dufraimont E, De Block C, Jacquemyn Y, Mekahli F, De Clippel K, Van Den Bruel A, Loccufier A, Laenen A, Minschart C, Devlieger R, Mathieu C. Characteristics and pregnancy outcomes across gestational diabetes mellitus subtypes based on insulin resistance. Diabetologia. 2019, 62(11):2118-2128.